# AN EXPERIMENTAL STUDY OF NEURAL NETWORKS FOR VARIABLE GRAPHS

**Xavier Bresson**[*]
School of Computer Science and Engineering
Nanyang Technological University, Singapore
xbresson@ntu.edu.sg

**Thomas Laurent**
Department of Mathematics
Loyola Marymount University
tlaurent@lmu.edu

## ABSTRACT

Graph-structured data such as social networks, functional brain networks, chemical molecules have brought the interest in generalizing deep learning techniques to graph domains. In this work, we propose an empirical study of neural networks for graphs with variable size and connectivity. We rigorously compare several graph recurrent neural networks (RNNs) and graph convolutional neural networks (ConvNets) to solve two fundamental and representative graph problems, subgraph matching and graph clustering. Numerical results show that graph ConvNets are 3-17% more accurate and 1.5-4x faster than graph RNNs. Interestingly, graph ConvNets are also 36% more accurate than non-learning (variational) techniques. The benefit of such study is to show that complex architectures like LSTM is not useful in the context of graph neural networks, but one should favour architectures with minimal inner structures, such as locality, weight sharing, index invariance, multi-scale, gates and residuality, to design efficient novel neural network models for applications like drugs design, genes analysis and particle physics.

## 1 INTRODUCTION

Originally introduced by Gori et al. (2005); Scarselli et al. (2009), several new graph neural network papers have been published in the recent years to solve graph learning problems in quantum chemistry by Duvenaud et al. (2015); Gilmer et al. (2017), natural language processing by Li et al. (2016); Marcheggiani & Titov (2017); Dauphin et al. (2017), and traffic control by Sukhbaatar et al. (2016). However, there has been no study that rigorously compares the two fundamental families of neural network architectures, ConvNets and RNNs, for graphs with variable size and connectivity. The main goal of this work is to provide a clear answer to which class of graph neural networks should be leveraged to design new learning models for graph problems with variable sizes.

The main contributions of this work are:

- In-depth experimental comparison of several graph RNNs and graph ConvNets.
- Design of analytically controlled experiments on two basic representative graph problems.
- Define the most effective graph learning architecture, i.e. ConvNet with edge gating mecanism and residuality.

## 2 NEURAL NETWORKS FOR GRAPHS WITH ARBITRARY SIZE

**Graph recurrent neural networks.** The earliest work of graph RNNs for arbitrary graphs was introduced by Gori et al. (2005); Scarselli et al. (2009). The authors proposed to use a vanilla RNN with multilayer perceptron to define a feature vector $h_i$ at vertex $i$:

$$h_i = f_{\text{G-VRNN}}(x_i, \{h_j : j \to i\}) = \sum_{j \to i} \mathcal{C}_{\text{G-VRNN}}(x_i, h_j) \tag{1}$$

where $\{j \to i\}$ is the set of neighbors $j$ pointing to $i$ (defined by the graph structure), $x_i$ is a data vector, $\mathcal{C}_{\text{G-VRNN}}(x_i, h_j) = A\sigma(B\sigma(Ux_i + Vh_j))$, $\sigma$ being the sigmoid function, and $A, B, U, V$ are

---

[*]XB is supported by NRF Fellowship NRFF2017-10.

the parameters to learn. Eq. (1) does not hold a closed-form solution and a fixed-point iterative scheme must be used.

Li et al. (2016) proposed to replace the multilayer perceptron in (1) by the GRU of Chung et al. (2014):

$$h_i = f_{\text{G-GRU}}\left(x_i, \{h_j : j \to i\}\right) = \mathcal{C}_{\text{G-GRU}}(x_i, \sum_{j \to i} h_j). \tag{2}$$

As Eq. (2) does not have an analytical solution, an iterative scheme was proposed: $h_i^{t+1} = \mathcal{C}_{\text{G-GRU}}(h_i^t, \bar{h}_i^t), \bar{h}_i^t = \sum_{j \to i} h_j^t, h_i^{t=0} = x_i$ and $\mathcal{C}_{\text{G-GRU}}(h_i^t, \bar{h}_i^t)$ is given by a standard GRU.

To complete the family of graph RNNs, we introduce the graph LSTM model by simply replacing the GRU gate by the LSTM gate into (2):

$$h_i = f_{\text{G-LSTM}}\left(x_i, \{h_j : j \to i\}\right) = \mathcal{C}_{\text{G-LSTM}}(x_i, h_i, \sum_{j \to i} h_j, c_i) \tag{3}$$

**Graph convolutional neural networks.** Sukhbaatar et al. (2016) introduced the first instantiation of graph ConvNets:

$$h_i^{\ell+1} = f_{\text{G-VCNN}}^\ell\left( h_i^\ell, \{h_j^\ell : j \to i\} \right) = \text{ReLU}\left(U^\ell h_i^\ell + V^\ell \sum_{j \to i} h_j^\ell\right), \tag{4}$$

where $\ell$ denotes the layer level, and ReLU is the rectified linear unit. We can refer to this architecture as the vanilla graph ConvNet. In the spirit of tree-LSTM proposed by Tai et al. (2015), Marcheggiani & Titov (2017) proposed to add a gating mechanism by replacing the neighborhood term $V^\ell \sum_{j \to i} h_j^\ell$ in (4) by $\sum_{j \to i} \eta_{ij} \odot V^\ell h_j^\ell$, where $\eta_{ij} = \sigma\left(A^\ell h_i^\ell + B^\ell h_j^\ell\right)$ is an edge gate.

We leverage the two models of Sukhbaatar et al. (2016); Marcheggiani & Titov (2017) and introduce a generic graph ConvNet architecture that benefits from gated edges and the complete graph topology from the previous layer $\ell$:

$$h_i^{\ell+1} = f_{\text{G-GCNN}}^\ell\left( h_i^\ell, \{h_j^\ell : j \to i\} \right) = \text{ReLU}\left(U^\ell h_i^\ell + \sum_{j \to i} \eta_{ij} \odot V^\ell h_j^\ell\right) \tag{5}$$

## 3 EXPERIMENTS

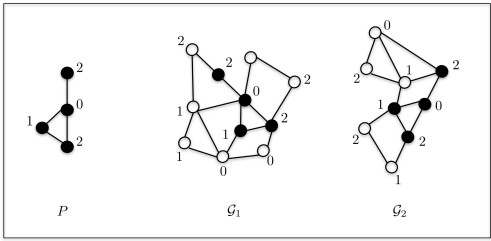 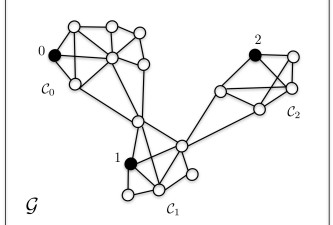

(a) Subgraph matching        (b) Semi-supervised clustering

**Subgraph matching.** We consider the subgraph matching problem presented by Scarselli et al. (2009), see Figure 1(a). The goal is to find the vertices of a given subgraph $P$ in larger graphs $G_k$ with variable sizes. Identifying similar patterns in different graphs is one of the most basic tasks for graph neural networks. All graphs are generated with a stochastic block model, see Abbe (2017). The subgraph $P$ has 20 nodes and the signal on $P$ is randomly generated from $\{0, 1, 2\}$. The size of larger graphs $G_k$ randomly varies between $150 - 250$ nodes with a signal randomly generated from $\{0, 1, 2\}$. Inputs of all neural networks are the graphs with variable size, and outputs are vertex classification vectors of input graphs (simply given by a fully connected layer from the hidden states).

All reported results are averaged over 5 trails. We run 5 algorithms; two graph RNNs with Gated Graph Neural Networks of Li et al. (2016) and the proposed graph LSTM in (3), and two graph

ConvNets with CommNets of Sukhbaatar et al. (2016), SyntacticNets of Marcheggiani & Titov (2017) and the proposed graph ConvNets in (5). We upgrade all existing models of Li et al. (2016); Sukhbaatar et al. (2016); Marcheggiani & Titov (2017) with a multi-layer version using residuality of He et al. (2016) for all architectures. Note that residuality significantly improved the original models of Li et al. (2016); Sukhbaatar et al. (2016); Marcheggiani & Titov (2017) by 10%. The learning rate and optimization scheme are optimized for each architecture individually. The loss is the cross-entropy and the accuracy is the number of nodes correctly classified.

Results are reported in Figure 1. The left plot compares the architectures w.r.t. the number $L$ of layers. The graph ConvNet models show a monotonous increase of performances with $L$, which is an expected and desired property. However, the graph RNN models see their performance decreases for large $L$. The middle plot reports the computational time for a batch of 100 training graphs. Graph RNN algorithms require more time to process the same number of graphs. Finally, the right plot shows the learning speed w.r.t. time.

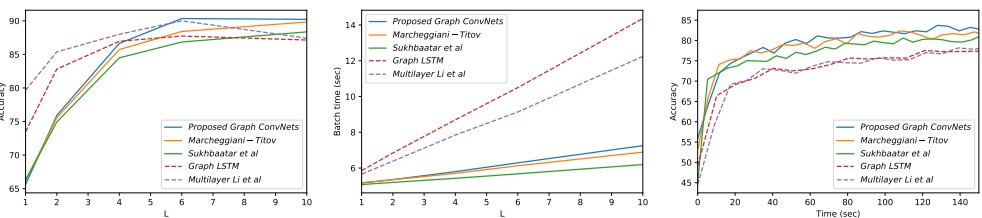

Figure 1: Subgraph matching.

**Semi-supervised clustering.** It is another fundamental and representative problem in graph theory. For this work, the problem consists in finding 10 communities given 1 single label for each community. The size of the graphs is randomly generated between 50 and 250 vertices. The same setting is used for all algorithms to provide the most fair comparison. Results are reported in Figure 2 and the same conclusions apply than the graph matching experience. Besides, the learning speed (right plot) of graph ConvNets is much faster than graph RNNs.

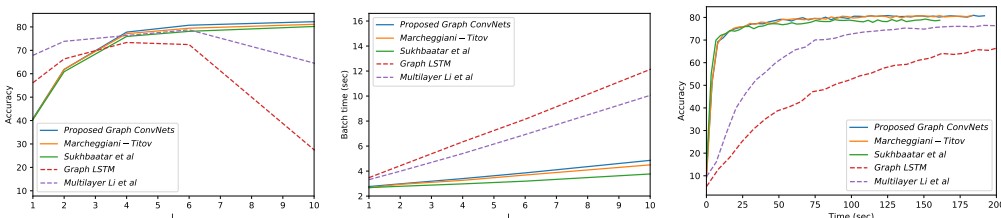

Figure 2: Semi-supervised clustering.

**Learning vs. non-learning techniques.** Finally, we compare the learning models to the variational/non-learning Dirichlet model for semi-supervised clustering proposed by Grady (2006). We run 100 experiments and report an average accuracy of 45.3% using the same setting whereas the performance of the best learning technique is 82%. The downside is the need to see 2000 training graphs to get to 82%. However, when training is done, the test complexity of these learning techniques is $O(E)$, where $E$ is the number of edges in the graph. This is an advantage over the variational Dirichlet model that requires to solve a sparse linear system of equations with complexity $O(E^{3/2})$, see Lipton et al. (1979).

**Summary of numerical results.**

- Graph ConvNets are 3-17% more accurate than graph RNNs.
- Graph ConvNets are 1.5-4x faster than graph RNNs.
- Graph ConvNets are 36% more accurate than variational (non-learning) techniques.
- The best architecture of graph ConvNets is equiped with gated edges and residuality.

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
