# OpenReview forum: "An Experimental Study of Neural Networks for Variable Graphs"
_ICLR.cc/2018/Workshop — Accept_

### Official Review · AnonReviewer1 · 2018-03-02
**interesting results**

**Rating:** 6
**Confidence:** 4

**Review:**

SUMMARY.

The paper presents an experimental study of several Graph Neural network architectures on two graph problems: subgraph matching and graph clustering.
The experiments are carried out on artificial data.


----------

OVERALL JUDGMENT
The paper is clearly written but it is quite dense.
There is no much novelty in the proposed approach but the experiments are well carried out, and the conclusions, although not very surprising are interesting.
Since this is a workshop paper I would suggest the authors focus more on the descriptions of architectural differences and maybe remove the paragraph about Learning vs. non-learning that seems a bit out of context.

----------

PROS
Good experimental setting
Interesting experimental results

CONS
The paper is too dense (Figures are too small)
Experiments only on artificial data

---

### Official Review · AnonReviewer2 · 2018-03-06
**Nice experimental comparison, not well-suited for the workshop**

**Rating:** 5
**Confidence:** 3

**Review:**

An experimental comparison of several graph neural networks w.r.t. subgraph matching and clustering is presented. The authors distinguish graph CNNs and graph RNNs. The result is that CNNs are faster and more accurate than RNNs.

In my opinion, authoritative experimental comparisons in this domain are very important. Although the authors claim to provide an "in-depth experimental comparison", my feeling is that this is not possible within the scope of a short paper. Many questions remain open, e.g., details on parameters and network architecture. Moreover, the relevance of the two tasks ist not clear.

Strong points:
  * Important subject.
  * Clear summary of the findings.

Weak points:
 * Due to the nature of an experimental study, the novelty and originality is limited.
 * Details of the experimental setup are missing.
 * Only two tasks/datasets.

In summary, I think that the article does not fit well into the format of a workshop paper and, therefore, cannot recommend its acceptance.

---

### Official Review · AnonReviewer3 · 2018-03-09
**Useful Insights**

**Rating:** 7
**Confidence:** 2

**Review:**

The paper demonstrates an empirical study to compare graph recurrent networks and convolutional networks in various applications. The results show a general advantage of graph convolutional networks.

The conclusion provides useful insights between comparisons between architectures for graphs. It certainly suffices as a workshop paper.

One problem is that the paper only selects a few possible instantiations for both recurrent and convolutional networks. There are a lot more ways to use either recurrent or convolutional networks for variable-size graphs, and the paper should include a discussion, even if experiments could not have be included for the workshop submission.

---

### Decision · Program_Chairs · 2018-03-20
**ICLR 2018 Workshop Acceptance Decision**

**Decision:**

Accept

**Comment:**

Congratulations, your paper was accepted to the ICLR workshop.

---

> ### Author Response · Authors · 2018-03-22
> **ICLR**
>
> We would like to thank the area chair and the referees for their time and the review of our paper.